# The Task at Hand: Fatigue-Associated Changes in Cortical Excitability During Writing

**DOI:** 10.3390/brainsci9120353

**Published:** 2019-12-02

**Authors:** Kezia T. M. Cinelli, Lara A. Green, Jayne M. Kalmar

**Affiliations:** Department of Kinesiology, Wilfrid Laurier University, Waterloo, ON N2L 3C5, Canada; cine2150@mylaurier.ca (K.T.M.C.); lgreen@wlu.ca (L.A.G.)

**Keywords:** corticospinal excitability, task-dependent, transcranial magnetic stimulation, fatigue, writing

## Abstract

Measures of corticospinal excitability (CSE) made via transcranial magnetic stimulation (TMS) depend on the task performed during stimulation. Our purpose was to determine whether fatigue-induced changes in CSE made during a conventional laboratory task (isometric finger abduction) reflect the changes measured during a natural motor task (writing). We assessed single-and paired-pulse motor evoked potentials (MEPs) recorded from the first dorsal interosseous (FDI) of 19 participants before and after a fatigue protocol (submaximal isometric contractions) on two randomized days. The fatigue protocol was identical on the two days, but the tasks used to assess CSE before and after fatigue differed. Specifically, MEPs were evoked during a writing task on one day and during isometric finger abduction to a low-level target that matched muscle activation during writing on the other day. There was greater variability in MEP amplitude (F (1,18) = 13.55, *p* < 0.01) during writing compared to abduction. When participants were divided into groups according to writing style (printers, *n* = 8; cursive writers, *n* = 8), a task x fatigue x style interaction was revealed for intracortical facilitation (F (1,14) = 9.90, *p* < 0.01), which increased by 28% after fatigue in printers but did not change in cursive writers nor during the abduction task. This study is the first to assess CSE during hand-writing. Our finding that fatigue-induced changes in intracortical facilitation depend on the motor task used during TMS, highlights the need to consider the task-dependent nature of CSE when applying results to movement outside of the laboratory.

## 1. Introduction

Single-pulse transcranial magnetic stimulation (TMS) elicits motor evoked potentials (MEPs) that are recorded from the muscle of interest using surface electromyography (EMG). MEPs are used as a measure of corticospinal excitability (CSE) that reflect the excitability of the pathway from the site of stimulation to the site of recording, such that both cortical and spinal mechanisms contribute to changes in the MEP evoked using single-pulse TMS. Intracortical mechanisms that may contribute to CSE are assessed using conditioned MEPs elicited via paired-pulse TMS. For example, paired-pulses with a brief interpulse interval (1–5 msec) provide a measure of short interval intracortical inhibition (SICI), whereas longer interpulse intervals (10–15 msec) provide a measure of intracortical facilitation (ICF) [1]. One challenge associated with using single- and paired-pulse TMS to assess CSE is the variability in MEP amplitude, both within and between participants. To minimize variability, most TMS protocols are conducted with the muscle at rest or during submaximal isometric muscle contractions to standardize levels of muscle activation and control for joint position, muscle length, movement, and other factors. Although these laboratory tasks are intended to minimize variability, they do not reflect activities of daily living to which the results may be extrapolated. This is problematic because CSE depends on the net excitability of the targeted brain region and net excitatory and inhibitory input to spinal motor neurons at the time of stimulation, all of which vary with different motor tasks and states [2,3,4]. Thus, the conclusions drawn from TMS studies depend on the motor task employed during stimulation and therefore may not translate to movement outside of laboratory settings.

Muscles of the hand (e.g., first dorsal interosseous (FDI) and adductor pollicis (AP)) are frequently used in studies of CSE because of the ease with which MEPs are elicited in distal muscles of the upper extremity. In such studies, MEPs are typically evoked with the hand at rest or during submaximal isometric contractions. For example, FDI MEPs are often assessed with the hand pronated with the wrist and hand secured to allow movement only at the metacarpal phalangeal joint of the index finger. Outside the lab, however, we use intrinsic muscles of the hands in a variety of positions, including power and precision grips. Measures of CSE differ between these grips. For example, FDI MEP amplitude is greater during conventional abduction tasks compared to power, pincer, or grasping tasks [5,6,7], whereas APB MEPs are greater during pincer tasks compared to during power tasks or at rest [8]. More direct measures made in the monkey reveal that corticospinal neurons are more active during a precision grip compared to a power grip, despite increased EMG activity in the latter [9]. This suggests that CSE is not simply related to the force of contraction, but also to the specific task. Pearce and Kidgell (2010) demonstrate that CSE is modulated when the precision required for a given task is increased [10] Furthermore, more complex tasks will require the involvement of proximal muscles for stability during finger movements. Flament and colleagues [11] compared index finger abduction to simple manual tasks including precision grip, power grip, grasping of a petri dish, and rotation of a bottle cap. Compared to isolated finger abduction, which was restricted to FDI use, all other tasks required activation of at least one additional muscle, which was speculated to contribute to the increased CSE found during the complex tasks [11].

Conventional laboratory tasks, such as matching force output to a static target displayed on a computer monitor, require very different cognitive demands compared to more natural motor tasks such as hand writing, use of a keyboard, or object manipulation. Nonetheless, there is a tendency to attribute the TMS results solely to the changes within or downstream to the primary motor cortex, and to overlook the influence of upstream cognitive processes transmitted to the primary motor cortex [12]. For example, internally guided movements, such as writing or drawing, have been shown to generate greater activation in the pre-supplementary motor area and dorsal premotor cortex as compared to externally guided movements, such as tracing [13]. Using the hand to convey language introduces additional cognitive influences on the motor system. In fact, mere observation of letters and words can alter CSE measured in the FDI when at rest [14,15]. Interestingly, this effect is specific to handwritten text, including handwritten “non-words”, and is not observed with typed text [14,15], indicating that the recognition of hand-writing represents a unique cognitive demand that is conveyed to the motor system, likely via the mirror neuron system [14]. Accordingly, it is unlikely that estimates of CSE obtained during simple, isometric laboratory tasks would translate directly to CSE during more complex tasks outside the lab. In this study, we compared a hand-writing task, which represents an internally generated and complex task that is familiar and relevant outside the laboratory, to a conventional externally guided, isometric force-matching task often used in TMS research.

We wanted to determine whether changes in CSE in response to a well-studied intervention (neuromuscular fatigue) would depend on the motor task that is used to produce background muscle activity during the delivery of TMS. TMS has been used for many years to understand the role of the central nervous system in neuromuscular fatigue (for review, see [16,17]). Such studies have revealed that CSE is briefly facilitated and then undergoes a more prolonged period of depression following fatigue when MEPs are evoked in resting muscles [18,19,20,21]. Post-exercise depression is also observed after a fatigue protocol when MEPs are evoked at rest just prior to a voluntary contraction [22]. On the other hand, MEPs recover much more quickly when assessed during a strong muscle contraction after fatigue [23,24], suggesting that reductions in CSE after fatigue are overcome with sufficient voluntary drive. Given that measures of CSE depend on limb position, muscle activation level, and other aspects of the motor task, such as different cognitive demands, we speculate that fatigue-induced changes in CSE made during a conventional laboratory task may not represent fatigue-induced changes in CSE made during the more relevant and complex task of writing. Therefore, the purpose of this study is to elicit neuromuscular fatigue and then compare fatigue-associated changes in CSE during writing to the same measures made during a conventional isometric finger abduction task. We hypothesized that CSE would be greater during the writing task compared to the simple isometric abduction task. Furthermore, we hypothesized that the effect of fatigue on CSE would depend on the motor task employed during stimulation, even when the fatigue task itself was the same. 

## 2. Materials and Methods

### 2.1. Participants

Twenty participants with a mean age of 22.6 ± 1.1 were recruited from the Wilfrid Laurier University student population. This sample size was based on a power calculation made using data from an earlier study of intracortical mechanisms of fatigue from our lab [22]. All participants were right-handed as determined by the Annett Handedness Questionnaire [25]. One participant was excluded from analysis due to an inability to complete the full testing protocol, leaving 19 participants (six male). Exclusion criteria included any neurological conditions, and orthopaedic conditions or pain of the hand, wrist, or arm. The study was approved by the Wilfrid Laurier University Research Ethics Board (REB #5381), and all participants provided written informed consent prior to participating.

### 2.2. Experimental Design

In this repeated-measures study design, each participant attended two testing sessions occurring no less than 48 h apart and was tested at the same time on both days to decrease between session variability [26]. On one day, participants completed a writing task and on the other day, participants completed an abduction task. The order of the two days was randomized and counterbalanced between participants. On each day participants began with a short familiarization to each task. Following familiarization, participants completed 90 pre-fatigue trials of the selected task (writing or abduction), a fatigue protocol of intermittent isometric abduction contractions, and finally 90 post-fatigue trials of the same task (writing or abduction) (Figure 1).

On each testing day participants completed a familiarization protocol including the writing task, the abduction task, and maximal voluntary contractions (MVCs). The writing task included 20 trials where the participant repetitively wrote the word “name” on a iPad tablet with a Adonit Pro3 precision disc stylus pen. The word “name” was specifically selected for the writing task because it is familiar, it is a short enough word to be written within the 5-s recording frame, and because the letters n, a, m, and e do not contain ascenders or descenders that would require greater finger movement. The tablet screen presented a blank 7 × 2 cm rectangle on a series of Powerpoint slides that were refreshed every 5 s. The rectangle served as the boundaries within which the participant was asked to write the word. These writing boundaries and the refreshing of the writing “page” on the tablet before each trial, allowed the participants to maintain a constant hand position and to minimize wrist deviations between trials. The participants were allowed to place the tablet in any position and orientation on the table in front of them. This position was marked and maintained between trials, blocks, and testing days. Participants were instructed to maintain the same self-selected grip of the pen (e.g., dynamic tripod, lateral tripod, quadrupod, etc.) for the duration of the protocol. In the abduction task, TMS was applied during isometric abduction of the right index finger against a force transducer for 3 s with a 2-s rest. The target level of contraction during the abduction task was set to the average RMS amplitude of FDI EMG activity over the 20 familiarization writing trials so that the contraction intensity was matched to activity required during writing for each participant (Figure 2). This level of activity was marked with horizontal cursors (±2.5%) to allow the participant to match it using a real-time smoothed and rectified EMG biofeedback channel. The location of the force transducer was adjusted to be in line with the self-selected graphic tablet location, such that shoulder and elbow angle were maintained between the writing and abduction tasks (Figure 3). The hand was in a pronated position for abduction with the third finger secured by a strap and wooden dowels placed on either side of the wrist. Participants then performed three isometric MVCs from which the highest was taken as maximal finger abduction force.

Following familiarization, the participant was set up for TMS. The pre-fatigue and post-fatigue tests included a total of 90 trials, which consisted of 30 pulses each of TMS stimulation type, including single-pulse test MEP (TS), and paired-pulse stimuli to assess short-interval intracortical inhibition (SICI), and intracortical facilitation (ICF). The 90 trials were performed in 3 blocks, with 10 sets per block and 3 trials per set to pseudo-randomize the TMS stimulation such that each set would include a single-pulse TS, SICI, and ICF evoked in random order (Figure 1). Each trial was 5 seconds in duration. On the writing day, the timing of the screen refresh was set to ensure that the stimulus was delivered as the participant wrote the letter “a” or “m”. Timing between pulses (5 s) was never changed. On the abduction day, the TMS stimulation occurred 2 seconds into the 5-second trial to ensure the participant was in the plateau portion of the submaximal, isometric, finger abduction contraction.

The fatigue protocol consisted of repeated 4-second isometric finger abduction contractions at 60% of the participant’s MVC, with two seconds between each contraction. Task failure was defined as the point at which force fell below 58% MVC for more than three seconds despite encouragement to meet and hold the 60% target. Immediately following task failure, participants completed a post-fatigue MVC before beginning the post-fatigue test (Figure 1). It is important to note that on both days, regardless of the condition (abduction or writing), participants performed the same isometric finger abduction fatigue protocol. In this way, we examined the task-dependent nature of TMS measures of CSE rather than the task-dependent nature of neuromuscular fatigue.

### 2.3. Experimental Set-Up and Recordings

#### 2.3.1. Electromyography and Force Recordings

The skin over the right FDI muscle was cleaned with isopropyl alcohol and two Ag/AgCl EMG electrodes were affixed in a bipolar configuration (0.5 cm recording surface, 1 cm interelectrode distance) over the muscle belly. A ground electrode was placed on the dorsal aspect of the right hand after being shaven and cleaned with alcohol. To allow the hand to rest comfortably on the graphic tablet, a glove covering the hand and the fourth and fifth digits was worn by participants. The skin over the extensor carpi radialis (ECR) and the flexor carpi radialis (FCR) was prepared for EMG by shaving, abrading, and cleansing with alcohol. Parallel bar surface EMG electrodes (10 × 1 mm Ag contacts, 1 cm interelectrode distance, DE-2.1 DELSYS Inc., Natick, MA, USA) were affixed over the muscle belly of the ECR and FCR, and a ground electrode was placed on the elbow. EMG and force signals were digitized at 2000 Hz using the Micro 1401-3 data acquisition unit and Signal 6.0 waveform acquisition software (Cambridge Electronics Design, Cambridge, UK). The FDI EMG signal was pre-amplified 300× and band-pass filtered from 15 to 450 Hz (Motion Lab Systems, Inc. Los Angeles, CA, USA). The ECR and FCR EMG data was amplified 1000× and band-pass filtered from 20 Hz to 450 Hz (Bagnoli-16, DELSYS Inc., Natick, MA, USA).

#### 2.3.2. Transcranial Magnetic Stimulation

Participants were seated at a table with an adjustable headrest situated in front of the forehead to allow the body and head to be supported in a comfortable writing position (Figure 3). A figure-eight magnetic stimulating coil (D70 coil, Magstim Company Ltd., Whitland, UK) was positioned over the primary motor cortex with the handle positioned posteriorly 45° to the midsagittal line and the induced current in a posterior to anterior direction and held in place using a lighting support arm and clamp (Manfrotto Supports, Cassola, Italy) with additional support and position maintenance by the investigator. The TMS coil was moved in small increments in order to determine the optimal site for generating a motor evoked potential (MEP) in the FDI via suprathreshold stimulations from the BiStim2 system (Magstim Company Ltd., Whitland, UK). Once located, this spot was marked on the cap worn by participants. Stimulator output was then adjusted to find the active motor threshold (AMT), determined as the minimum intensity that elicited a 200 μV MEP in 5 out of 10 of trials while the participant maintained a target matched to the average EMG over 20 writing trials recorded during the familiarization period at the beginning of each experimental day. Pulses were delivered at intervals of at least 5 s during threshold hunting. Paired-pulse TMS was used to assess SICI and ICF with a conditioning stimulus of 80% AMT preceding the test MEP set at 120% AMT. To elicit SICI, the conditioning stimulus preceded the test MEP by 3 ms To elicit ICF, the conditioning stimulus preceded the test MEP by 12 ms [27].

#### 2.3.3. Motor Evoked Potentials (MEPs)

Motor evoked potential (MEP) peak-to-peak amplitudes were measured offline using Signal 6.0 (Cambridge Electronics Design, Cambridge, UK). Trials were excluded if (a) the participant responded incorrectly (e.g., no contraction), (b) TMS stimulation occurred between EMG bursts during writing, and (c) no MEP was elicited (MEP < 200 μV) [28]. Exclusion criteria a, b, and c were based on trial by trial inspection of the waveform data. SICI and ICF MEPs were normalized to the corresponding test MEP within the same set (each series of three randomized trials that include TS, SICI, and ICF). If a test MEP was excluded, MEPs were normalized to the test MEP in the next closest set such that the conditioned MEP was always normalized to an unconditioned MEP that was no further than three trials away. Test MEP peak-to-peak amplitudes are reported in millivolts (mV). The coefficient of variation of the test MEP amplitude was calculated for each participant to determine the consistency of the measure. Conditioned MEP peak-to-peak amplitudes (SICI and ICF) were normalized to the test MEP amplitude and are therefore reported as ratio values where <1 would indicate inhibition and >1 would indicate facilitation. Sample data from one participant are shown in Figure 4.

#### 2.3.4. Cortical Silent Period

The duration of the test MEP CSP was measured offline using a custom script in MATLAB (MathWorks, Inc., Natick, MA, USA). CSP duration was calculated from the point of the test pulse stimulus delivery to the point at which EMG activity returned to prestimulus activity (i.e., average RMS amplitude calculated over a 300-ms period prior to stimulation). CSP was calculated as the average for the 30 test pulse trials only (CSP following paired-pulses were not included).

### 2.4. Statistical Analysis

Analysis was performed using Statistica 13.2 (TIBCO Software Inc., Palo Alto, CA, USA). For TMS pulses, trial outliers were removed if they fell >2 standard deviations from the average within each individual participant. Two participants were excluded from the SICI analysis due to a large number of outliers occurring within a single block of trials. One participant was removed from the CSP analysis due to an inability return to the target level of EMG activity following stimulation. Assumptions for the analysis of variance (ANOVA) were tested for each variable using the Shapiro–Wilk test for normality. Test MEP amplitudes were not normally distributed, thus a 1/sqrt transformation was applied for statistical analysis. Preliminary analysis was performed to ensure that there was no effect of block (i.e., the 3 blocks of 30 trials pre- and post-fatigue). No block main effect was present for any variable, therefore all pre-fatigue trials and all post-fatigue trials were averaged for each participant. 

To examine potential differences between testing days a dependent samples *t*-test was conducted on baseline measures (see Table 1). To determine the effect of task on dependent measures, a one-way repeated measures ANOVA was performed comparing pre-fatigue values between writing and the abduction task. To determine the effect of task on the dependent measures during fatigue, a 2 (task: abduction versus writing) × 2 (fatigue: pre versus post) repeated measures ANOVA was performed for each dependent variable. The number of participants included in each analysis is noted in Table 2 and Table 3.

As the study progressed, we found that of the 19 participants, 8 wrote in cursive, 8 printed, and 3 used a combination of cursive and printing. Accordingly, we conducted an additional post hoc analysis of writing style. This separate-groups post hoc analysis included writing style (cursive vs. printing) as a categorical predictor to determine the baseline effect (task × writing style) and the effect of task and style on fatigue (task × fatigue × writing style) in a posteriori analysis. The three individuals who used a combination of printing and cursive writing strategies to complete the task were excluded from this analysis. Because this analysis of writing style was not planned a priori, it is important to note that it is underpowered.

Tukey’s HSD testing was used for post hoc analysis where applicable. Partial eta-squared was calculated in Statistica for the ANOVA models. Cohen’s d was calculated for the pre-fatigue to post-fatigue values where applicable [29].

## 3. Results

### 3.1. Baseline Measures

The baseline measures recorded during familiarization (force and FDI RMS amplitude during MVCs, FDI activity of writing, and AMT) were not different between the abduction and writing days. Additionally, the number of fatiguing trials performed was not different between the two days. The amount of fatigue elicited was not significantly different (Task × Style: F (1,13) = 0.88, *p* = 0.37) between printers on the writing day (−26.0% MVC force), cursive writers on the writing day (−18.6%), printers on the abduction day (−27.5%), and cursive writers on the abduction day (−22.6%).

### 3.2. Effect of Task

#### Motor Evoked Potentials

In the pre-fatigue trials, there was no effect of task (writing versus abduction) on the amplitude of the test MEP (F(1,18) = 0.60, *p* = 0.45, $\eta_{p}^{2}$ = 0.03, Figure 5A), ICF (F(1,18) = 1.69, *p* = 0.21,$\;\eta_{p}^{2}$= 0.09, Figure 5C), or CSP (F(1,17) = 1.14, *p* = 0.30, $\eta_{p}^{2}$= 0.06). However, the coefficient of variation of test MEP amplitude was 23% greater in the writing task compared to the abduction task (F(1,18) = 13.55, *p* < 0.01, $\eta_{p}^{2}$ = 0.43). The coefficient of variation of the test MEP was higher in both the printing (48.4 ± 7.3, *n* = 8) and cursive writing (42.4 ± 5.5, *n* = 8) styles, as compared to abduction in printers (38.3 ± 7.9, *n* = 8) and abduction in cursive writers (35.1 ± 8.2, *n* = 8). Although there appeared to be a trend toward a greater level of inhibition (SICI) during writing compared to abduction (F(1,16) = 4.40, *p* = 0.052, $\eta_{p}^{2}$= 0.22, Table 2, Figure 5B), this analysis included only 17 participants and was not adequately powered. When writing style (printing vs. cursive writing) were factored in (task × style: F(1,13) = 8.00, *p* = 0.01, $\eta_{p}^{2}$= 0.38, Table 3), post hoc analysis revealed a significant difference (*p* < 0.01) in SICI between printing (0.71 ± 0.13, *n* = 7) and abduction (1.03 ± 0.28, *n* = 7, Figure 6B) in printers, but no difference between cursive writing (0.86 ± 0.26, *n* = 8) and abduction (0.87 ± 0.17, *n* = 8) in cursive writers.

### 3.3. Muscle Activity

Task × fatigue repeated-measures ANOVAs were conducted for FDI, FCR, and ECR surface EMG measures of muscle activation. Level of FDI muscle activation was successfully maintained between the two tasks (F(1,18) = 0.60, *p* = 0.45, $\eta_{p}^{2}$ = 0.03). This was expected because average writing RMS amplitude was used as the muscle activation target during the abduction task. The abduction task elicited greater levels of FCR activity (F(1,18) = 7.24, *p* = 0.02, $\eta_{p}^{2}$ = 0.29) and lower levels of ECR activity (F(1,18) = 7.73, *p* = 0.01, $\eta_{p}^{2}$= 0.30) compared to the writing task. Analysis of writing styles (task × style) revealed a significant interaction for both FCR activity (F(1,14) = 4.70, *p* = 0.048, $\eta_{p}^{2}$= 0.25; Figure 7A) and ECR activity (F(1,14) = 5.25, *p* = 0.04, $\eta_{p}^{2}$= 0.27; Figure 7B). Post hoc analysis identified that cursive writing required significantly less FCR activity (12.58 ± 7.42 µV) than abduction (29.79 ± 17.57 µV) in the cursive writing group. There was no significant difference between FCR activity during printing (18.36 ± 9.10 µV) or abduction (20.92 ± 13.07 µV) in the printing group. In the extensor muscle, post hoc analysis identified that printing (154.22 ± 71.53 µV) required significantly more ECR activity than abduction (70.36 ± 47.40 µV) in the printers. There was no significant difference between ECR activity during cursive writing (99.07 ± 27.63 µV) or abduction (92.84 ± 39.92 µV) in the cursive group.

### 3.4. Effect of Task on Fatigue

#### 3.4.1. Motor Evoked Potentials

The task × fatigue repeated-measures ANOVA was not significant for any measure of CSE (see Table 2 for test MEP, SICI, and ICF results), including test MEP coefficient of variation (F(1,18) = 3.59, *p* = 0.07, $\eta_{p}^{2}$ = 0.17) and CSP (F(1,17) = 0.29, *p* = 0.60, $\eta_{p}^{2}$ = 0.02). However, when writing style was taken into consideration, a 3-way (task × fatigue × style) repeated-measures ANOVA revealed a significant 3-way interaction for ICF (F(1,14) = 9.90, *p* < 0.01, $\eta_{p}^{2}$ = 0.41) (Table 3). Post hoc analysis revealed a significant increase in ICF from pre-fatigue to post-fatigue (28.1%, d = 0.94, *p* = 0.04) in the printing group. Planned comparisons were conducted using percent change.

Scores (pre- to post-fatigue) to determine the task × style effect on fatigue. The increase in facilitation in printing (28.1%) was significantly different than the decrease in facilitation of printers performing the abduction task (−12.0%, *p* < 0.01, Figure 6C). There were no significant effects of writing style on the test MEP or SICI (see Table 3), test MEP coefficient of variation (F(1,14) = 0.02, *p* = 0.88, $\eta_{p}^{2}$ = 0.002), or CSP (F(1,13) = 0.30, *p* = 0.60, $\eta_{p}^{2\;}$ = 0.02).

#### 3.4.2. Muscle Activity

The task × fatigue repeated measures ANOVA was not significant for FDI activity (F(1,18) = 1.31, *p* = 0.27, $\eta_{p}^{2}\;$ = 0.07), FCR activity (F(1,18) = 1.69, *p* = 0.21,$\;\;\eta_{p}^{2}$ = 0.09), or ECR activity (F(1,18) = 0.86, *p* = 0.37, $\eta_{p}^{2}$ = 0.05). Similarly, an analysis of writing style (task × fatigue × style) did not yield any significant interactions for FDI activity (F(1,14) = 0.57, *p* = 0.46, $\eta_{p}^{2}$ = 0.04), FCR activity (F(1,14) = 0.57, *p* = 0.47, $\eta_{p}^{2}\;$= 0.04), or ECR activity (F(1,14) = 2.82, *p* = 0.12, $\eta_{p}^{2}$ = 0.17).

## 4. Discussion

Our study demonstrates that fatigue-associated changes in CSE depend on the motor task completed during single- and paired-pulse TMS stimulation. Not only did we find differences in fatigue-associated changes in cortical excitability assessed during hand-writing compared to isometric finger abduction, but we also found differences between participants who completed the writing task by printing compared to those who wrote in cursive. Specifically, we found that the same fatigue protocol elicited an increase in ICF during printing, but not during cursive writing. It is important to note that the fatigue protocol itself was the same on each day, and only the motor task during pre-fatigue and post-fatigue assessments of CSE differed. Furthermore, because the level of FDI muscle activation was held constant during the two tasks, the differences in CSE between writing and abduction days were not simply due to differences in voluntary drive to that muscle.

The present study examined CSE during voluntary movement. Because the level of muscle activation during voluntary movement affects levels of CSE [30,31,32], it was critical in the present study to match FDI EMG between tasks. We accomplished this by measuring FDI muscle activation during the writing task in a familiarization period each day, and then using biofeedback to set a target on the abduction day that matched muscle activation during writing for each participant. This approach to matching muscle activation between the two tasks (writing and abduction) was equally effective in printers and cursive writers. Therefore, differences in intracortical measures of excitability pre- and post-fatigue cannot be attributed to differing FDI EMG activity between the tasks.

During baseline (pre-fatigue) testing, the variability of single-pulse MEP amplitudes was greater in the writing task than the abduction task. This was expected because the abduction task is a single-joint, externally cued task frequently used in TMS studies for the purpose of reducing variability. The abduction task was maintained at a stable force level using visual feedback. To make the writing task more functionally-relevant, we did not control the style, or speed of each participant’s writing. Therefore, it is not surprising that the writing task resulted in greater variability than the abduction task, both within and between subjects. Similar results have been found comparing test–retest reliability of MEP amplitudes between static and dynamic tasks in the lower limb [33], where MEPs evoked during a static task (i.e., plateau at a target force level) were less variable than those evoked during a dynamic task (i.e., continuously increasing force level). The style of writing did not impact the MEP amplitude variability, as we found higher coefficient of variation values for both printing and cursive writing compared to abduction. A future challenge in TMS research will be to utilize laboratory tasks that balance variability with the relevance of the task to motor activities outside of the lab.

We hypothesized that CSE would be greater during writing compared to abduction due to the complexity of the writing task. However, single-pulse MEP amplitude was not different between tasks. Reports of differences in CSE between precision tasks and conventional abduction tasks are inconsistent. Original research in monkeys suggested that corticospinal neurons were more active during a precision task compared to a power task [9], a finding supported by several studies that found greater MEP amplitudes during complex, precision tasks compared to simple, power tasks [4,11,34]. Alternatively, larger MEP amplitudes have been reported during conventional abduction tasks compared to power, pincer, or grasping tasks [5,6,7]. The discrepancy between our findings and previous research may be that other studies have used precision tasks that were visually guided and externally controlled, whereas our writing task is a dynamic, internally generated task that is retrieved and implemented from memory [35,36]. Furthermore, writing involves higher levels of activation in the dorsal premotor cortex in comparison to simple finger contraction tasks, as well as unique activation in multiple brain regions (e.g., premotor cortex and anterior putamen) not activated during tapping or “zigzagging” finger actions [37]. This association of writing with higher cognitive demands would suggest that writing is a more complex task in comparison to simple finger contractions. On the other hand, because writing is learned and practiced from a young age into adulthood, it is also associated with a degree of automaticity [38]. Accordingly, our finding that writing was not associated with increased CSE compared to abduction, may have been due to either the increased complexity or the automaticity of writing compared to unpracticed, less natural, and externally guided precision-grip tasks used in previous studies.

Despite the fact that baseline (pre-fatigue) CSE did not differ between tasks, levels of intracortical inhibition (SICI) assessed using paired-pulse TMS trended (*p* = 0.052) toward greater inhibition during writing compared to abduction. When participants were subdivided into those who printed and those who wrote in cursive (excluding those who used a combination of printing and cursive within a single word), printers had the greatest intracortical inhibition during the writing task. We hypothesize that this may be due to the increased control required during the more intermittent task of writing distinct letters during printing, compared to the continuous nature of cursive writing (Figure 2). The differences in intracortical inhibition observed between writing styles may also be explained by the activity of the proximal forearm muscles. While FDI EMG activity between abduction and writing did not differ, extensor (ECR) activity was greater and flexor (FCR) activity was lower during the writing task compared to abduction. It has been suggested that activity and position of proximal muscles can have an effect on the corticospinal pathway leading to the distal muscle of interest [39,40,41]. This relationship has been shown between the FDI and proximal arm muscles (including the ECR, FCR, and deltoid muscles), where proximal muscle activity resulted in facilitation of distal muscle MEPs [39,40,41]. Similarly, forearm position (pronation vs. semi-supinated) is also known to alter CSE [42]. In our study, however, the forearm was semi-supinated for both printing and cursive writing, and therefore forearm position does not explain the differences in intracortical excitability that we observed when we compared the two different writing styles. The role that proximal muscle activation may play in the CSE of distal muscles highlights the importance of choosing laboratory tasks that more closely resemble natural movements outside of the laboratory to which we aim to extrapolate our results. 

TMS is used to assess changes in corticospinal and intracortical excitability in response to many different interventions and perturbations (e.g., fatigue, strength training, skill training, disease, injury, aging, and pharmacological agents). We sought to determine whether changes in CSE following an intervention depend on the motor task used to elicit muscle activity during the delivery of TMS. To this end, we employed neuromuscular fatigue as an acute intervention, and measured changes in CSE during two different motor tasks before and after two identical fatigue protocols. It is well established that neuromuscular fatigue is associated with changes in CSE, specifically a reduction in unconditioned MEP (test MEP) amplitudes in the target muscle following neuromuscular fatigue [43,44]. This depression has been attributed to central mechanisms of fatigue [44,45]. Paired-pulse TMS techniques have been used to identify intracortical mechanisms of fatigue [22,46,47]. Following fatiguing contractions, SICI decreased [18,46,48]; however, the association between fatigue and ICF is not as clear. For example, ICF measured in the biceps brachii decreases during a sustained fatiguing contraction [48,49], whereas ICF measured in the FDI is elevated at the point of task failure when assessed at rest [22,46]. In other studies, ICF does not change with fatigue [47,49]. It has been suggested that decreased intracortical inhibition [18] and increased facilitation following fatigue [22] may serve as a compensatory mechanism to optimize motor output as fatigue develops.

In our study, there was no task-dependent effect of fatigue on any of the corticospinal and intracortical excitability measures between writing and abduction. However, most of our participants used one of two distinct writing styles: printing or cursive. Therefore, we completed additional a posteriori analyses to determine whether there was an effect of writing styles on fatigue-induced changes in CSE, and found that printers had a significant increase in ICF following fatigue. This is consistent with previous reports of increased ICF following fatigue that employ more conventional laboratory tasks [46,50]. However, this only occurred during printing, and was not found during cursive writing or the conventional finger abduction task. Previous research has suggested that increased ICF post-fatigue may be a mechanism to compensate for peripheral contractile failure or reduced upstream drive to the primary motor cortex [22]. However, this would not explain the increased ICF seen in printing but not in cursive writing or abduction. Possibly the differences in wrist stabilization during printing and cursive, reflected by differences in wrist extensor and wrist flexor muscle activity during the two tasks, had a greater impact on ICF than on SICI. Furthermore, it is possible that printing employs cortical circuits that are affected by isometric finger abduction fatigue task differently than the cortical circuits employed in cursive writing. Although there is very little research regarding the neural control of printing versus cursive writing, one clinical study reports impairment in cursive writing, but not printing, drawing, or the ability to draw continuous loops, following ischemic damage to the parietal lobe [51]. Another clinical case found that the ability to write in cursive was lost with the development of a large cranial tumor impinging on the left frontal lobe, while the ability to print remained intact. In this case, the patient regained the ability to write in cursive following tumor resection [52]. Case studies like these support the idea that printing and cursive writing have different cortical representations, and may therefore be associated with different inputs to the primary motor cortex. This novel finding certainly warrants further exploration, as printing and cursive writing appear to be two different tasks that further demonstrate the task-dependent nature of fatigue-associated changes in CSE.

We started this repeated-measures study with 20 participants based on a power analysis conducted using data from a previous study from our lab that also investigated intracortical mechanisms of fatigue [22]. Of the 20 participants we recruited, 1 participant was unable to complete the study, and 2 participants were excluded from the SICI analysis. Accordingly, this analysis is not robustly powered, and these data should be interpreted cautiously. The analysis of hand writing style was not planned in advance. From the start, we told participants to write as they preferred given that our aim was to assess corticospinal excitability during a natural motor task. As the study progressed, we found that many university students were unable to write in cursive, and we conducted the post hoc analysis of writing style. It should be noted that this a posteriori analysis of writing style (cursive vs. printing) is underpowered. Furthermore, in the present study, we assessed only three intracortical mechanisms (SICI, ICF, and cSP). Each of these measures has been associated with different types of neurotransmission (SICI with GABA-A receptors, ICF with glutamatergic transmission, and the cSP with GABA-B receptors) as reviewed by Reis [1]. It is possible that other intracortical (e.g., long interval intracortical inhibition) and interhemispheric pathways (e.g., interhemispheric inhibition) may reveal fatigue-associated changes that are dependent on the motor task that is conducted during stimulation. This is an area that requires further investigation

## 5. Conclusions

This study is the first to assess CSE and the effect of fatigue during a writing task. Although fatigue-associated changes in CSE have been well studied over the last three decades, this study highlights the importance of considering the task used during TMS measures of corticospinal and intracortical excitability. Although controlled laboratory tasks are required to reduce variability of motor evoked potentials to allow for reproducible results, it is important to note that CSE is task-dependent. Because of this, measures of CSE made during a laboratory task may not translate to motor tasks outside of the lab. Despite the fact that the fatigue task and the level of FDI muscle activation during TMS was the same in our study, hand-writing revealed fatigue-associated changes in CSE that were not evident during abduction. Therefore, the task-dependent nature of CSE emphasizes the need for experimental paradigms that better reflect relevant motor tasks or at least acknowledge the differences between the task employed in the laboratory and the movements outside of the lab to which results may be extrapolated. With this in mind, it is essential that studies of CSE consider the “task at hand”.

## Figures and Tables

**Figure 1 brainsci-09-00353-f001:**
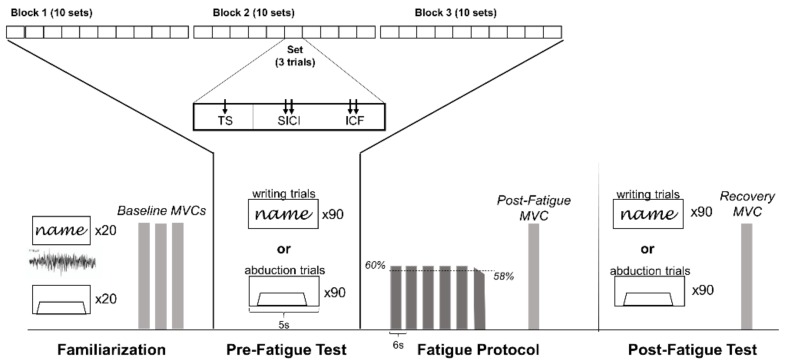
Each day began with a familiarization period that included 20 repetitions of each task to record muscle activation during writing, familiarizing participants with the force-tracing task, and an assessment of baseline maximal voluntary contraction (MVC). Participants then completed the Pre-Fatigue Test, which included 90 trials (abduction or writing task), followed by a Fatigue Protocol that was comprised of submaximal (60% MVC target) abduction contractions until failure (force fell below 58% MVC force for >3 s during attempts to hold the 60% target), followed by a post-fatigue MVC. The protocol ended with the Post-Fatigue Test that was identical to the Pre-Fatigue Test followed by a recovery MVC. The 90 pre-fatigue and post-fatigue trials were separated into 3 blocks of 10 sets each. Each set contained 3 randomized trials: (1) test MEP (TS), (2) short-interval intracortical inhibition (SICI) paired-pulse, and (3) intracortical facilitation (ICF) paired-pulse stimulation, resulting in pseudo-randomization of TS, SICI, and ICF across each block.

**Figure 2 brainsci-09-00353-f002:**
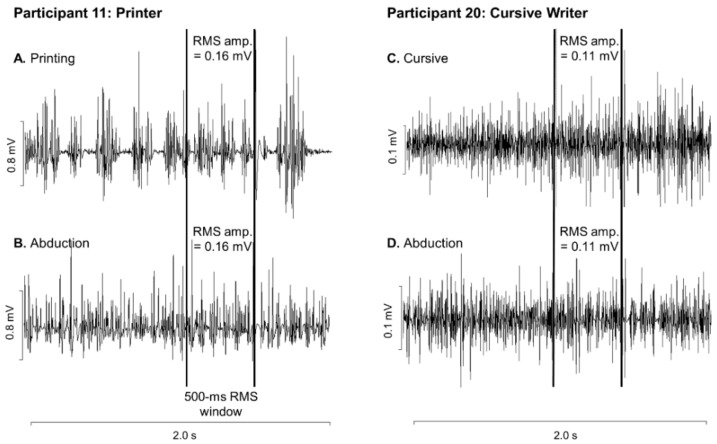
First dorsal interosseous (FDI) electromyographic activity in two representative participants: one who printed (**A**,**B**) and one who wrote in cursive (**C**,**D**). FDI EMG activity is shown during writing (**A,C**) and abduction (**B,D**). The vertical lines denote the 500-msec window prior to TMS stimulation from which the root-mean-square activity was calculated.

**Figure 3 brainsci-09-00353-f003:**
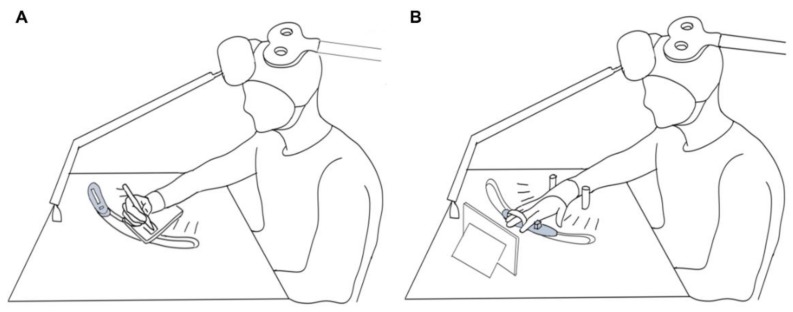
Experimental set-up of participant positioning during the writing (**A**) and abduction (**B**) tasks. The table design allowed the position of the force transducer to match the self-selected writing position of the graphic tablet.

**Figure 4 brainsci-09-00353-f004:**
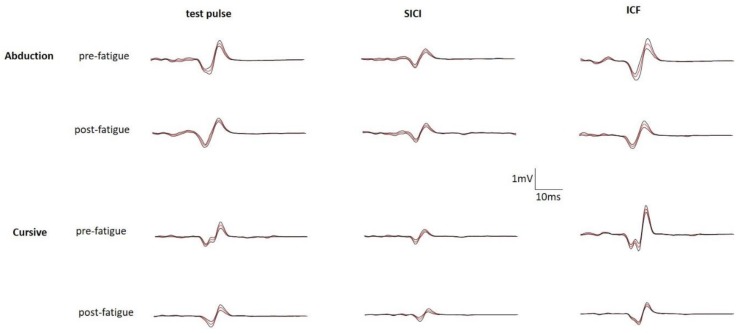
Block averages (red line) and error (black lines) of evoked potentials for one participant. Each block included 10 single test pulses, 10 SICI paired-pulses, and 10 ICF paired-pulses. This participant completed the writing task in cursive. Pre-fatigue averages are shown above the post-fatigue averages.

**Figure 5 brainsci-09-00353-f005:**
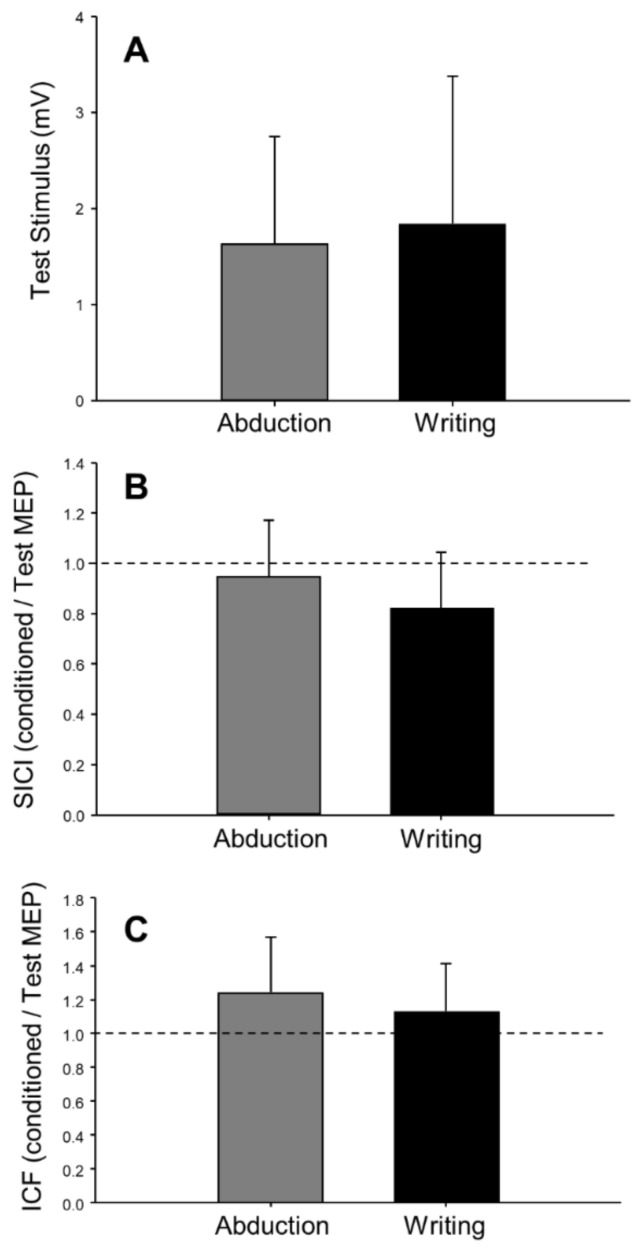
Pre-fatigue levels of test MEP (*n* = 19), intracortical facilitation (*n* = 19), and intracortical inhibition (*n* = 17) during finger abduction and writing (*n* = 19). There were no differences between test stimulus (**A**), short-interval intracortical inhibition (SICI; **B**), and intracortical facilitation (ICF; **C**) between the two tasks. In panels B and C, dashed lines represent no effect of the conditioning stimulus. Values below the line represent inhibition (conditioned MEP < test MEP), and values above the line represent facilitation (conditioned MEP > test MEP). Error bars represent standard deviation.

**Figure 6 brainsci-09-00353-f006:**
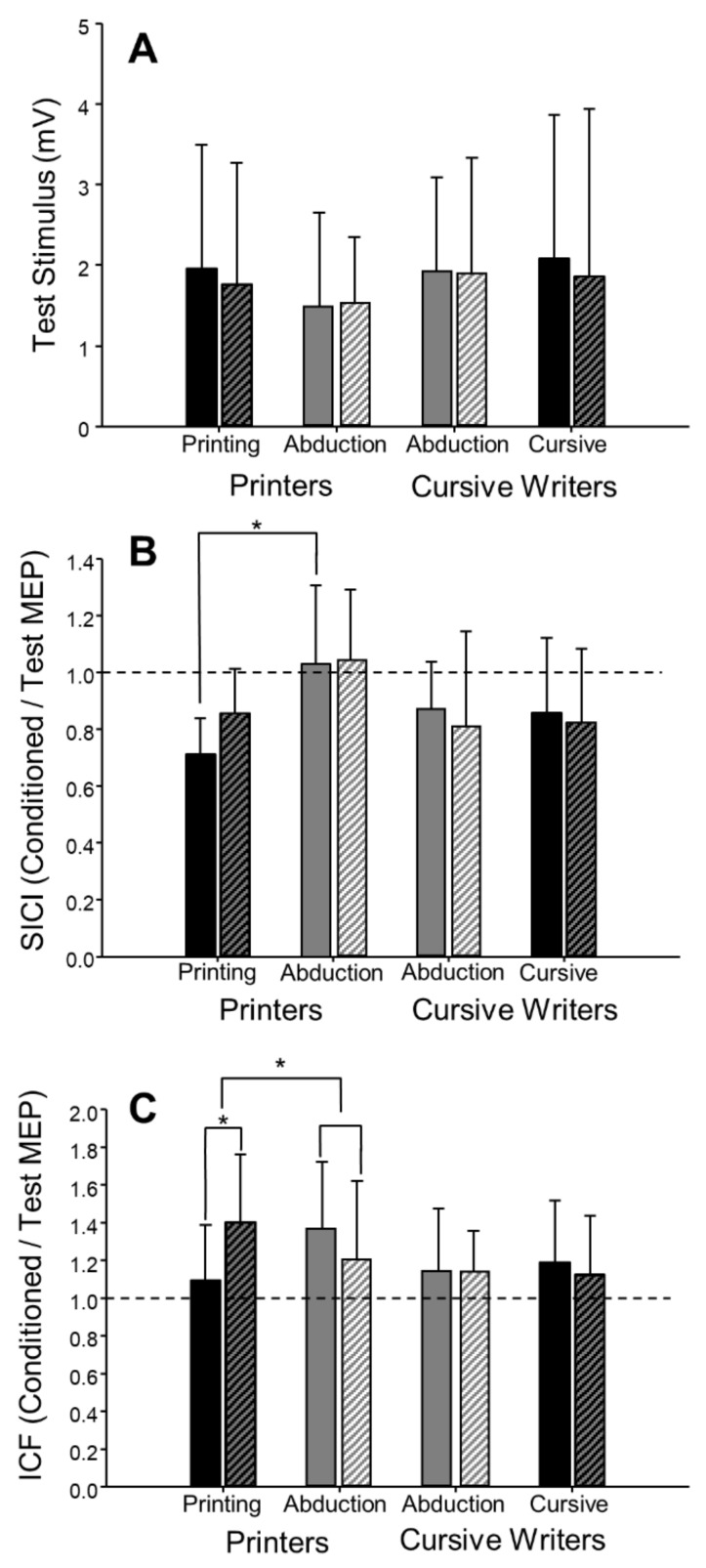
Motor evoked potentials during finger abduction and writing in printers (*n* = 8) and cursive writers (*n* = 8). There were no differences in single-pulse (test stimulus) MEPs between writers and printers or between tasks (test stimulus; **A**). Short-interval intracortical inhibition (SICI; **B**) was greater during printing than during abduction pre-fatigue (*, *p* < 0.01). In printers, there was a fatigue-associated increase in intracortical facilitation (ICF; **C**) when assessed during printing, but not during abduction (*, *p* < 0.01). Solid bars represent pre-fatigue values, and hatched bars represent post-fatigue values. In panels B and C, dashed lines represent no effect of the conditioning stimulus. Values below the line represent inhibition (conditioned MEP < test MEP), and values above the line represent facilitation (conditioned MEP > test MEP). Error bars represent standard deviation.

**Figure 7 brainsci-09-00353-f007:**
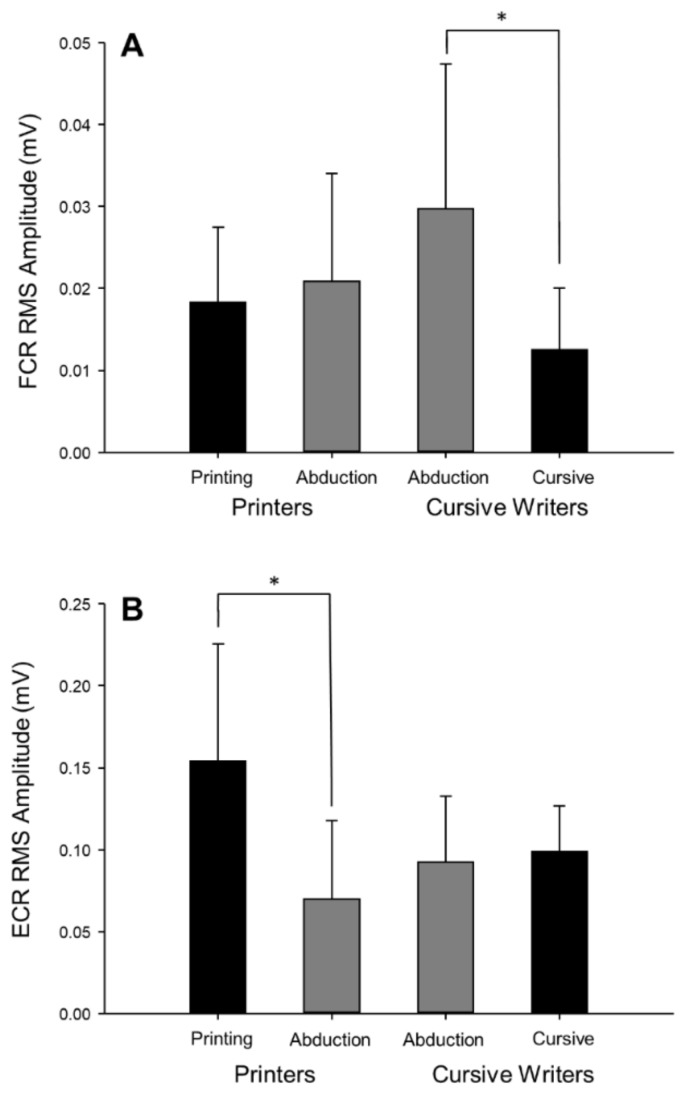
Pre-fatigue levels of wrist flexor (**A**) and wrist extensor (**B**) activity during the abduction and writing tasks in printers and cursive writers. Wrist flexor muscle activity was higher in the writing task than in the abduction task in cursive writers, whereas wrist extensor muscle activity was higher in the writing task than in the abduction task in the printers (*, *p* < 0.05). This analysis included participants who completed the writing task using a pure printing (*n* = 8) or cursive (*n* = 8) strategy. Error bars represent standard deviation.

**Table 1 brainsci-09-00353-t001:** Baseline measures made during the familiarization period on the abduction and writing days. These measures included maximal index finger abduction force and first dorsal interosseous (FDI) activity during MVCs, the level of FDI activity during 20 writing trials, the number of fatiguing sets completed, and active motor threshold (AMT) for TMS (Mean ± SD).

	Abduction Day	Writing Day
Maximal Force (*N*)	19.49 ± 5.80	24.05 ± 11.38
Maximal FDI RMS amplitude (mV)	0.91 ± 0.38	0.94 ± 0.40
Writing EMG (% max)	18.7 ± 6.88	20.5 ± 9.62
AMT (%MSO)	39.1 ± 8.20	41.0 ± 7.30
# of Fatiguing Trials	43.50 ± 26.34	48.83 ± 45.34

FDI: first dorsal interosseous; MVC: maximal voluntary contraction; AMT: active motor threshold; RMS: root mean squared; EMG: electromyography; MSO: maximum stimulator output.

**Table 2 brainsci-09-00353-t002:** Motor evoked potentials recorded from the FDI muscle and results of the repeated measures ANOVAs comparing Task (abduction versus writing) and Task × Fatigue conducted with all 19 participants.

	Abduction Task			Writing Task			Effects	
Measure	Pre-Fatigue	Post-Fatigue	% Change Effect Size	Pre-Ftg	Post-Ftg	% Change Effect Size	*F* Value ^§^ (Task)	*F* Value (Task × Fatigue)
**Test MEP** **(*n* = 19)**	1.64 ± 1.11	1.54 ± 1.13	−5.9% *d* = 0.09	1.85 ± 1.53	1.64 ± 1.65	−11.2% *d* = 0.13	F(1,18) = 0.60, *p* = 0.45, η_p_^2^ = 0.03	F(1,18) = 2.22, *p* = 0.15, η_p_^2^ = 0.11
**SICI** **(*n* = 17)**	0.95 ± 0.22	0.94 ± 0.30	−0.7% *d* = 0.03	0.82 ± 0.22	0.86 ± 0.21	4.1% *d* = 0.16	F(1,16) = 4.40, *p* = 0.052, η_p_^2^ = 0.22	F(1,16) = 0.36, *p* = 0.56, η_p_^2^ = 0.02
**ICF** **(*n* = 19)**	1.25 ± 0.33	1.19 ± 0.30	−4.2% *d* = 0.16	1.13 ± 0.28	1.26 ± 0.33	11% *d* = 0.41	F(1,18) = 1.69, *p* = 0.21, η_p_^2^ = 0.09	F(1,18) = 3.60, *p* = 0.07, η_p_^2^ = 0.17

^§^ Calculated on the pre-fatigued values only. % Change is calculated as ((post-pre)/pre) × 100 from the unrounded data. Cohen’s *d* is calculated for the pre- to post-fatigue means. Abbreviations: SICI: short-interval intracortical inhibition; ICF: intracortical facilitation.

**Table 3 brainsci-09-00353-t003:** Results of the ANOVA comparing Task × Style (printing versus cursive) and Task × Fatigue × Style conducted with the subset of participants (*n* = 16) that could be classified as printers (bold font) or cursive writers.

		Abduction Task			Writing Task			Effects	
Measure	Method (*n*)	Pre-Fatigue	Post-Fatigue	% Change Effect Size	Pre-Fatigue	Post-Fatigue	% Change Effect Size	*F* Value ^§^ (Task × Style)	*F* Value (Task × Fatigue × Style)
**Test MEP**	**Printers (8)**	**1.49 ± 1.16**	**1.54 ± 0.82**	**3.0%** ***d* = 0.05**	**1.96 ± 1.54**	**1.77 ± 1.50**	−10.0% *d* = 0.13	F(1,14) = 2.95, *p* = 0.11, η_p_^2^ = 0.17	F(1,14) = 0.77, *p* = 0.40, η_p_^2^ = 0.05
	Cursive writers (8)	1.93 ± 1.16	1.90 ± 1.43	−1.4% *d* = 0.02	2.09 ± 1.78	1.86 ± 2.08	−10.9% *d* = 0.12		
**SICI**	**Printers (7)**	**1.03 ± 0.28**	**1.05 ± 0.25**	**1.5%** ***d* = 0.06**	**0.71 ± 0.13**	**0.86 ± 0.16**	20.4% *d* = 1.02	F_(1,13)_ = 8.00, *p* = 0.01, η_p_^2^ = 0.38	F(1,13) = 0.56, *p* = 0.47, η_p_^2^ =0.04
	Cursive writers (8)	0.87 ± 0.17	0.81 ± 0.33	−6.6% *d* = 0.22	0.86 ± 0.26	0.82 ± 0.26	−4.1% *d* = 0.13		
**ICF**	**Printers (8)**	**1.37 ± 0.35**	**1.21 ± 0.41**	**−12.0%** ***d* = 0.43**	**1.09 ± 0.29**	**1.40 ± 0.36**	28.1% *d* = 0.94	F(1,14) = 2.71, *p* = 0.12, η_p_^2^ = 0.16	F(1,14) = 9.90, *p* = 0.007, η_p_^2^ =0.41
	Cursive Writers (8)	1.15 ± 0.33	1.14 ± 0.21	−0.4% *d* = 0.02	1.19 ± 0.33	1.12 ± 0.31	−5.6% *d* = 0.21		

^§^ Calculated on the pre-fatigued values only. % Change is the (post − pre)/pre fatigue values. Cohen’s *d* is calculated for the pre- to post-fatigue means. Abbreviations: SICI: short-interval intracortical inhibition; ICF: intracortical facilitation.

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
