# Peer review of "The Task at Hand: Fatigue-Associated Changes in Cortical Excitability During Writing"

_brainsci, 2019, doi:10.3390/brainsci9120353_

Round 1
Reviewer 1 Report
[General comment]
This experiment compared the effect of the fatigue on the MEP size, SICI, and ICF between the tonic abduction task and writing task. In my impression, there are many important methodological concerns on this study.
[Novelty of the study]
This experiment compared the effect of the fatigue on the MEP size, SICI, and ICF between the tonic abduction task and writing task. We know that for the same amount of the voluntary contraction level, the corticospinal excitability is dependent on the task (e.g. power grip, precision grip; Hasegwa et al. 2001). Moreover, the MEP amplitude in the FDI during power grip, tip pinch, grasping a dish, rotating a cap of a bottle is greater than the simple index finger abduction (Flament et al. 1993). As shown these previous findings, the task dependency of the MEP has been elucidated. Accordingly, the novelty of the present study is not so great.
[Task dependency]
The tonic abduction task and writing task have too many aspects of the difference. During abduction task in this study, only one muscle contracts, but during writing, multiple intrinsic and extrinsic hand muscles are active. Thus, many factors, such as enslaving effect (Zatisiorsky et al. 1998, 2000) or surround inhibition (Beck et al. 2008, 2011) must be present during writing. Moreover, writing is complex cognitive task, but finger abduction is not. Such difference in cognitive demand also affect the MEP. Finally, in the abduction task, the participants abducted the index finger with visual feedback of the EMG level. This force adjustment with visual feedback must differentiate the process of the index finger abduction task from the process of writing task. Thus, the authors can not obtain any conclusive evidence from the comparison between the tasks. To conduct an experiment scientifically, the two tasks must involve only one aspect of the difference.
[EMG recording]
In this study, high cut filter of the EMG was as low as 450 Hz, and sampling rate was 2kHz. We know that the MEP is very sharp wave, which makes negative slope in several ms and makes positive slope in several ms. Thus, high cut filter level has to be higher than 3kHz, and sampling rate has to be 5 kHz or higher (e.g., Aoki et al. 2019, Scientific Report). Otherwise, the sharp MEP wave is partially lost when recording. In addition, to record the surface EMG from the FDI, the placement of the electrodes based on the belly-tendon montage must be better than placing both the anode and cathode over the muscle belly.
[Coil direction]
Information regarding coil orientation is missing in this manuscript. The coil direction is very important information of the experimental design using the TMS. The elicited types of the interneurons are different among the coil directions (Ni et al. 2011, J Neurophysiol; Sakai et al. 1997, Exp Brain Res).
[AMT]
The fluctuation of the EMG trace greater than 50 micro V in a specific time gate was considered to be present of the MEP to determine the active motor threshold (AMT). This is inappropriate methodology. To determine the AMT, voluntary contraction of the tested muscle must be done. Voluntary contraction causes background EMG possibly more than 50 micro V. Thus, the fluctuation of the EMG trace greater than 50 micro V of the amplitude may be the background EMG. In order to avoid such mistake, the MEP in the FDI is usually defined to be the response whose amplitude is greater than 200 micro V to determine the AMT (Ni et al. 2009, J Neurophysiol).
[Test TMS intensity]
In this study, the test stimulus intensity was 120% of AMT and the conditioning stimulus intensity was 80% of AMT. This does not work. The waves (D and I waves) induced by the TMS is dependent on the TMS intensity (Di Lazzaro et al. 1998, Electroenceph clin Neurophysiol). Thus, the influence of the conditioning TMS is greatly dependent on the test MEP size (e.g., Pinto et al. 2001, Exp Brain Res). Because of this reason, the test stimulus intensity of the test TMS is usually determined based on the MEP amplitude (Daskalakis et al. 2004, J Physiol).
[TMS timing]
Writing proceeds with some phases. Thus, the timing of the TMS during writing is important. However, in this study, the timing of the TMS is not cared. This is a great weak point in this study.
Author Response
Thank you for the time you have taken to review our manuscript. I have responded to your comments and queries in-line below. Your comments are in bold, and our responses are in regular font.
This experiment compared the effect of the fatigue on the MEP size, SICI, and ICF between the tonic abduction task and writing task. In my impression, there are many important methodological concerns on this study.
[Novelty of the study]
This experiment compared the effect of the fatigue on the MEP size, SICI, and ICF between the tonic abduction task and writing task. We know that for the same amount of the voluntary contraction level, the corticospinal excitability is dependent on the task (e.g. power grip, precision grip; Hasegwa et al. 2001). Moreover, the MEP amplitude in the FDI during power grip, tip pinch, grasping a dish, rotating a cap of a bottle is greater than the simple index finger abduction (Flament et al. 1993). As shown these previous findings, the task dependency of the MEP has been elucidated. Accordingly, the novelty of the present study is not so great.
Although it is well established that corticospinal (CSE) is dependent on voluntary contraction level and task, researchers continue to assess CSE during laboratory tasks that do not reflect natural motor tasks outside the lab. While it is significant that MEP amplitude differs between power and pinch grip, and grasping a dish, and rotating a bottle cap, none of these laboratory tasks reflect the complex natural task of hand writing. Given that this the first study to have assessed corticospinal excitability during hand writing, we do feel that this study is novel. In addition, this is the first study to use two different motor tasks during TMS stimulation after exactly the same motor task to elicit fatigue. In so doing, this is the first study to establish that fatigue-associated changes in intracortical excitability depend on the task used during stimulation rather than the task used to fatigue the muscle. This is novel and significant because it addresses highlights the problems associated with extrapolating findings from simple isometric, single-joint operating systems to complex motor tasks outside the laboratory.
[Task dependency]
The tonic abduction task and writing task have too many aspects of the difference. During abduction task in this study, only one muscle contracts, but during writing, multiple intrinsic and extrinsic hand muscles are active. Thus, many factors, such as enslaving effect (Zatisiorsky et al. 1998, 2000) or surround inhibition (Beck et al. 2008, 2011) must be present during writing. Moreover, writing is complex cognitive task, but finger abduction is not. Such difference in cognitive demand also affect the MEP. Finally, in the abduction task, the participants abducted the index finger with visual feedback of the EMG level. This force adjustment with visual feedback must differentiate the process of the index finger abduction task from the process of writing task. Thus, the authors can not obtain any conclusive evidence from the comparison between the tasks. To conduct an experiment scientifically, the two tasks must involve only one aspect of the difference.
Yes, we completely agree that isometric abduction and the natural writing tasks are completely different in many ways. By directly comparing the two in this study, we provide evidence that a simple laboratory task does not yield the same results as a natural motor task. This was our primary objective and the purpose of the study. This study was not designed or intended to determine the factors that contribute to the differences between these tasks. We did control body and arm position with a custom-built apparatus. We also recorded forearm muscle activity because the use of a single joint operating system in conventional laboratory tasks negates the important role that proximal muscle activation may play in setting corticospinal gain. Finally, we did control for muscle activation of the muscle of interest (FDI). In short, this scientific experiment was designed to meet the a priori objectives and test the a priori hypotheses described in the manuscript introduction.
[EMG recording]
In this study, high cut filter of the EMG was as low as 450 Hz, and sampling rate was 2kHz. We know that the MEP is very sharp wave, which makes negative slope in several ms and makes positive slope in several ms. Thus, high cut filter level has to be higher than 3kHz, and sampling rate has to be 5 kHz or higher (e.g., Aoki et al. 2019, Scientific Report). Otherwise, the sharp MEP wave is partially lost when recording. In addition, to record the surface EMG from the FDI, the placement of the electrodes based on the belly-tendon montage must be better than placing both the anode and cathode over the muscle belly.
Thank you very much for bringing this recent paper to my attention. I apologize, but I cannot find an article with the first author Aoki, in Scientific Reports in 2019. Perhaps it is in press? I would like to read the article, and would very much appreciate the full citation. My sincere apologies if I am missing it, but I cannot seem to find it using the terms you have provided.
I have been conducting TMS experiments since 1996, and have never collected TMS data 5000 Hz. In fact, I note that many researchers do not collect at such a high frequency (e.g. Gagné, M., & Schneider, C. (2007). Brain Research; Opie, G. M., Ridding, M.C., Semmler, J. G. (2014). J Neurophysiol; Gordon, C.L., Spivey M. J., Balasubramaniam, R. (2017). Neuroscience Letters; Darling, W. G., Wolf, S. L., & Butler, A. J. (2006). Experimental Brain Research; Kouchtir-devanne, N., Capaday, C., Cassim, F., Derambure, P., & Devanne, H. (2018). J. Neurophysiol.; Bunday, K. L., Tazoe, T., Rothwell, J. C., & Perez, M. A. (2014). Journal of Neuroscience; Hunter, S.K., McNeil, C.J., Butler, J.E., Gandevia, S.C., Taylor, J.L. (2016). Exp Brain Res). Furthermore, Darling et al, (2006) state, "We examined whether sampling at 1,000 Hz was sufficient by recording MEPs in one subject at 5,000 Hz. Regression analysis of peak-to-peak amplitudes (PPamp) of MEPs in the 5,000 Hz sampled data on PPamp from the same data down-sampled to 1,000 Hz (i.e., every 5th point) was used. Correlation coefficients exceeded 0.99 and regression slopes were near 1.0, showing that the 1,000 Hz sampling rate was sufficient to obtain PPamp of EDC MEPs.”
Like many others, we have used a bipolar EMG electrode arrangement that has yielded high-quality EMG recordings from the first dorsal interosseous muscle (see Figure 4, added in response to another reviewer’s comment).
[Coil direction]
Information regarding coil orientation is missing in this manuscript. The coil direction is very important information of the experimental design using the TMS. The elicited types of the interneurons are different among the coil directions (Ni et al. 2011, J Neurophysiol; Sakai et al. 1997, Exp Brain Res).
Yes, the coil orientation should certainly have been included. Thank you for catching this omission. It has been added to lines 218-219.
[AMT]
The fluctuation of the EMG trace greater than 50 micro V in a specific time gate was considered to be present of the MEP to determine the active motor threshold (AMT). This is inappropriate methodology. To determine the AMT, voluntary contraction of the tested muscle must be done. Voluntary contraction causes background EMG possibly more than 50 micro V. Thus, the fluctuation of the EMG trace greater than 50 micro V of the amplitude may be the background EMG. In order to avoid such mistake, the MEP in the FDI is usually defined to be the response whose amplitude is greater than 200 micro V to determine the AMT (Ni et al. 2009, J Neurophysiol).
Yes, you are correct. Thank you very much for noting this error. We have mistakenly included the criteria for resting motor threshold in the methods (which we use in other studies). In this study, however, we assessed active motor threshold. This description should have read, “AMT was the minimum stimulator output that induced MEPs of more than 200 μV in at least 5 out of 10 consecutive trials during isometric contractions that were matched to the RMS amplitude during hand writing. As indicated in Table 1, this equates to approximately 20% MVC, a value used to calculate AMT in other papers (e.g. Ni et al., J. Physiol, 583(Pt 3): 971–982). We have corrected this error in lines 226-228.
[Test TMS intensity]
In this study, the test stimulus intensity was 120% of AMT and the conditioning stimulus intensity was 80% of AMT. This does not work. The waves (D and I waves) induced by the TMS is dependent on the TMS intensity (Di Lazzaro et al. 1998, Electroenceph clin Neurophysiol). Thus, the influence of the conditioning TMS is greatly dependent on the test MEP size (e.g., Pinto et al. 2001, Exp Brain Res). Because of this reason, the test stimulus intensity of the test TMS is usually determined based on the MEP amplitude (Daskalakis et al. 2004, J Physiol).
There are two approaches to setting test stimulus intensities (Rossini et al., Clin Neurophysiol. 2015; 126(6): 1071–1107). Test stimulus amplitude can be set to a given MEP amplitude (e.g. 1-mV), but it can also be set relative to the AMT or RMT. There are advantages to both approaches. Using a set MEP amplitude (such as 1-mV) reduces between participant variability in test MEPs. However, a test MEP set to a given amplitude (e.g. 1-mV) may be in the toe region of the stimulus response curve in one participant, and in the steep linear region of the stimulus response curve for another participant. It has been argued that use of a test stimulus intensity based on MEP amplitude is problematic for this reason.
“Indeed, the 1-mV method is irrational since 1 mV will correspond to differing points on the recruitment curve for different subjects.” Kukke et al., J Clin Neurophysiol. 2014; 31(3): 246–252).
Using a relative test stimulus intensity (e.g. 120% MT) is more likely to place participants in the same region of the stimulus response curve. A more moderate view would be that both methods are acceptable, each with advantages and limitations. Like a great many others, we have elected to use the relative approach.
[TMS timing]
Writing proceeds with some phases. Thus, the timing of the TMS during writing is important. However, in this study, the timing of the TMS is not cared. This is a great weak point in this study.
I apologize, but I am not sure what you mean by, “…timing of TMS is not cared…”. We agree that timing is very important. As noted in 173-178, timing was tightly controlled. Using frame-based software, we delivered pulses (single or paired) every 5-s. This was true in both the abduction and writing conditions. In the writing condition, the tablet screen (which displayed the command to begin writing) was timed to refresh such that the TMS pulse was delivered on the letters “a” or “m” during the 5-s frame. Any trials that fell between these two letters (between EMG bursts) were excluded. Accordingly, in the writing trials, all stimuli were delivered at exactly 5-s intervals and all MEPs were elicited during either the “a” or the “me” of the written word.
Reviewer 2 Report
The authors present novel research comparing MEPs following two-different types of motor tasks (writing vs isometric finger abduction) that induced fatigue. Further, sub-analyses showed that an interaction effect was present between those who printed versus those who used cursive writing.
This is an interesting piece of research which certainly has merit, particularly in cautioning translating findings of fatigue research generally. However, the study is underpowered somewhat thereby conclusions should be tempered.
Major
Why only SICI and not LICI? Particularly in that cSP was measured and while measuring slightly different parameters of inhibition GABAa and GABAb (Hanajima and Ugawa) many studies include SICI and LICI (as it is not too onerous to take a few more measures). Need to include why LICI was not included and state this as a limitation.
Can you explain your sample size? Given SICI was "trending" it is difficult to accept that only ICF showed differences post intervention.
A limitations of the research needs to be included in the discussion.
Minor:
Lines 35 to 38. It would be useful to naive readers to illustrate what is meant by SICI and ICF, i.e. 1-5 ms for SICI and 10-15 ms for LICI.
Lines 61-62. The Flament study is a classic but suggest also including more recent research that shows CSE is altered in similar movement tasks requiring greater precision (illustrating task-dependent) modulation of CSE and cSP - see Pearce and Kidgell J Sci Med Sport. 2009;12:280-283 and Pearce and Kidgell J Sci Med Sport. 2010;13:167-171.
Line 106. Could the authors justify their sample size calculation? Please include years after 22.6±1.1
Lines 218-223. More details on threshold determination and tonic contraction of the FDI are required. For example were the pulses delivered monophonic or biphasic (as this can influence MT determination and MEP amplitudes). See Chipchase et al Clin Neurophysiol. 2012;123(9):1698-1704.
Further, was a maximal voluntary contraction (MVC) determined and if so what percentage of MVC did participants hold during threshold determination? (Low-level could mean 2%, 5% or 10% of MVC all of which could influence MEP characteristics during testing). Relating to this, depending upon the intensity of low-level contraction, 50 microvolts is very low for a minimum during active (50 microvolts is usually the benchmark for resting MT and 200 microvolts for active MT due to EMG patterns depending upon level of contraction). This appears to be the case in line 227 where MEPs <200 microvolts are excluded (contradicting the 50 microvolt threshold). Finally, what time between each pulse delivered needs to be included to illustrate that paired-pulse measures where not influenced by too-quick application of pulses. See Hanajima R, and Ugawa Y. The Oxford Handbook of Transcranial Stimulation. New York: Oxford University Press; 2008:103-117.
Lines 398-401. Whilst I appreciate illustrating the reference to discuss MEP variability, the paper by Van Hedal et al was in the lower limb (Tib Ant). You should consider including citations that present MEP variability in an upper limb muscle, preferably an intrinsic hand muscle, to better illustrate you point here.
Lines 425-426. Relating to my point regarding underpowered findings, it is difficult to accept the word "trending" when the sample size is relatively small (and you had one person's data excluded).
Lines 425-445. It would be useful to discuss GABA-ergic activity explaining SICI differences but also it would be helpful to discuss why you didn't find cSP changes, given both SICI and cSP are derived from GABAb receptor activity.
Author Response
Thank you for the time you have taken to review our manuscript. I have responded to your comments and queries in-line below. Your comments are in bold, and our responses are in regular font.
The authors present novel research comparing MEPs following two-different types of motor tasks (writing vs isometric finger abduction) that induced fatigue. Further, sub-analyses showed that an interaction effect was present between those who printed versus those who used cursive writing.
This is an interesting piece of research which certainly has merit, particularly in cautioning translating findings of fatigue research generally. However, the study is underpowered somewhat thereby conclusions should be tempered.
Thank you very much for your comments and suggestions. We completely agree that while we designed the experiment to test our a priori hypotheses with a large sample (20 participants) based on a power calculation from an earlier experiment (Sharples et al., 2016), the power was compromised by the fact that the sample was reduced to 19 with one participant dropping out, and further reduced to 17 participants for the analysis of SICI. The unanticipated a posteriori analysis we conducted on hand writing style (printing vs cursive) is underpowered. We fully agree with your recommendation to temper our conclusions (described more fully below).
Major
Why only SICI and not LICI? Particularly in that cSP was measured and while measuring slightly different parameters of inhibition GABAa and GABAb (Hanajima and Ugawa) many studies include SICI and LICI (as it is not too onerous to take a few more measures). Need to include why LICI was not included and state this as a limitation.
One of the challenges of assessing cortical contributions to fatigue is that fatigue is transient. While post-fatigue TMS measures are made, participants are recovering from fatigue. Fitting these measures into a short period of time is therefore essential. Our aim was not to determine what cortical mechanisms contribute to fatigue. Certainly, if that had been our aim, then we would fully agree that it would have been necessary to assess all possible intracortical mechanisms (as well as CMEPs and M waves). However, our purpose was to demonstrate that TMS estimates of fatigue-induced changes in corticospinal excitability depend on the task completed during stimulation and that such estimates (and therefore the conclusions drawn from such measures) would differ if recorded during a natural motor task. In order to assess each measure that we selected with adequate repeats (30 trials per variable) and still allow for 5-s between stimuli to prevent unintentional conditioning, we limited our variables to three commonly reported parameters (test, ICF, and SICI).
We have included a suggestion for future work in lines 518-524. Here, we suggest that our finding that fatigue-associated changes in intracortical excitability depend on the task employed during TMS may extend to other known contributions to neuromuscular fatigue,
Can you explain your sample size? Given SICI was "trending" it is difficult to accept that only ICF showed differences post intervention.
We started with 20 participants in this repeated-measures study. We based this on a power analysis conducted using data from a previous study from our lab (Sharples et al., PLoS One, 2016) which indicated that we would need 15 participants to detect fatigue-induced differences in SICI, and 20 participants to detect differences in ICF (with α=0.05, β=0.08). The analysis of hand writing style (printing vs writing) was not planned in advance. As the study progressed, we found that many university students were unable to write in cursive. From the start, we told participants to write as they preferred given that our aim was to assess corticospinal excitability during a natural motor task. When we found that there were equal numbers of participants who chose to write purely in cursive or purely via printing, we conducted the post hoc analysis. It is possible that with increased sample size, we may have not have observed fatigue-associated changes in SICI, given that neither the means, the p value, nor the effect size suggested a robust difference.
A limitations of the research needs to be included in the discussion.
We have added a section that discusses sample size and statistical power as a limitation to this post hoc analysis, as well as an acknowledgement that there are other intracortical and interhemispheric pathways that were not explored in this study. This has been added to lines 509 to 518.
Minor:
Lines 35 to 38. It would be useful to naive readers to illustrate what is meant by SICI and ICF, i.e. 1-5 ms for SICI and 10-15 ms for LICI.
We have added in the range of interpulse intervals used to assess SICI and ICF. The citation at the end of that sentence also provides an excellent and very thorough review of these paradigms (line 38).
Lines 61-62. The Flament study is a classic but suggest also including more recent research that shows CSE is altered in similar movement tasks requiring greater precision (illustrating task-dependent) modulation of CSE and cSP - see Pearce and Kidgell J Sci Med Sport. 2009;12:280-283 and Pearce and Kidgell J Sci Med Sport. 2010;13:167-171.
We referenced the Flament study because it demonstrates the activation of additional muscles with a complex task compared to a simple task (this was the point we were making in this sentence). While the Pearce and Kidgell studies do present some interesting findings with regards to task-dependent modulation of CSE, they do not present activity from muscles other than the FDI and would therefore not be relevant citations to support this point. However, we have included a reference to Pearce and Kidgell (2010) in the preceding sentence (lines 61-62) to support the notion that the degree of precision required also results in the modulation of CSE .
Line 106. Could the authors justify their sample size calculation? Please include years after 22.6±1.1
We have added a note that the sample size was based on a power calculation from Sharples et al., PLoS One, 2016 (as described a preceding comment).
Lines 218-223. More details on threshold determination and tonic contraction of the FDI are required. For example were the pulses delivered monophonic or biphasic (as this can influence MT determination and MEP amplitudes). See Chipchase et al Clin Neurophysiol. 2012;123(9):1698-1704.
We have added more details about the contraction intensity used to assess AMT in lines 226-228. TMS was delivered using a Magstim Bistim2 TMS unit via a D70 (double 70mm) coil as noted in lines 216-217. This system, which is very commonly used in TMS studies of fatigue, delivers a monophasic pulse.
Further, was a maximal voluntary contraction (MVC) determined and if so what percentage of MVC did participants hold during threshold determination? (Low-level could mean 2%, 5% or 10% of MVC all of which could influence MEP characteristics during testing). Relating to this, depending upon the intensity of low-level contraction, 50 microvolts is very low for a minimum during active (50 microvolts is usually the benchmark for resting MT and 200 microvolts for active MT due to EMG patterns depending upon level of contraction). This appears to be the case in line 227 where MEPs <200 microvolts are excluded (contradicting the 50 microvolt threshold). Finally, what time between each pulse delivered needs to be included to illustrate that paired-pulse measures where not influenced by too-quick application of pulses. See Hanajima R, and Ugawa Y. The Oxford Handbook of Transcranial Stimulation. New York: Oxford University Press; 2008:103-117.
The procedures used to identify AMT were not correctly described in the manuscript. The criteria for resting motor threshold were mistaken included in the methods. In this study, however, we assessed active motor threshold. AMT was the minimum stimulator output that induced MEPs of more than 200 μV in at least 5 out of 10 consecutive trials during isometric contractions that were matched to the RMS amplitude during hand writing trials conducted during the familiarization phase. As indicated in Table 1, this equates to approximately 20% MVC, a value used to calculate AMT in other papers (e.g. Ni et al., J. Physiol, 583(Pt 3): 971–982). We have corrected this error and provided additional clarification in lines 225-228.
Pulses were delivered at least 5-s apart during threshold hunting (this was accomplished using a 5-s frame duration).
Lines 398-401. Whilst I appreciate illustrating the reference to discuss MEP variability, the paper by Van Hedal et al was in the lower limb (Tib Ant). You should consider including citations that present MEP variability in an upper limb muscle, preferably an intrinsic hand muscle, to better illustrate you point here.
Thank you. Yes, that is a good point – we have not made a direct comparison to an upper limb muscle, and it is important to make this clear. We have revised this sentence to clearly indicate that this comparison is made to a lower limb muscle. We have not found a paper that directly compares variability of MEP amplitudes between static and dynamic tasks in the upper extremity.
Lines 425-426. Relating to my point regarding underpowered findings, it is difficult to accept the word "trending" when the sample size is relatively small (and you had one person's data excluded).
This reference to a trend pertained a sample of 17 participants. Given that this sample size was based on an a priori power calculation that predicted that a sample size of 15 participants would be sufficient to detect fatigue-induced changes in SICI, it seemed reasonable to refer to a p value of 0.052 as a trend. However, we do agree that this must be approached with caution given the variability in our results. We have clarified that this analysis was underpowered in lines 309-312, and included a section at the end of the Discussion (lines 509-518) that reiterates power as a limitation in this study.
Lines 425-445. It would be useful to discuss GABA-ergic activity explaining SICI differences but also it would be helpful to discuss why you didn't find cSP changes, given both SICI and cSP are derived from GABAb receptor activity.
SICI is associated with GABA-A activity (Ziemann et al. 1996; Ilic et al. 2002 ), whereas LICI (Werhahn et al. 1999; McDonnell et al. 2006) and cSP (Siebner et al. 1998; Werhahn et al. 1999) are associated with GABA-B activity. We agree that discussion of potential GABAergic contributions to our measures of inhibition would be helpful and have added to this to lines 518-521. Because you and other reviewers have suggested that we do not overstate our SICI findings, we have not included an in-depth discussion of the neurotransmitter receptors that may contribute to these intracortical circuits.
Reviewer 3 Report
The task at hand: Fatigue-associated changes in 2 cortical excitability during writing Authors: Kezia T. M. Cinelli, Lara A. Green and Jayne M. Kalmar
Summary: This study examined corticospinal excitability during hand-writing and contrasted this with a common laboratory task before and after fatigue. The authors report that fatigue-induced changes in intracortical facilitation are task dependent, which has implications for studying corticospinal excitability during functional movements outside of the laboratory.
General comments:
Overall, this is a well conducted study with findings that are potentially relevant to those studying cortical excitability and motor control. However, there are a few statistical items that should be clarified. First, given that the actual sample size is 8 (having split the handwriting style into 2 groups), was an a priori power calculation performed? If not, why not? Next, although the authors provide partial eta squared and Cohen’s d, these are not easily interpreted here with such a small sample size. This makes some of the finding uncertain (see specific comments below). Furthermore, with ANOVA with repeated measures, the authors should comment on whether sphericity was tested. If sphericity was violated, an appropriate adjustment needs to be employed (e.g., Greenhouse Geisser correction). For some findings this may change whether the differences reported are significant (see specific comments below) or not.
Specific comments:
Abstract
Lines 17-18: There was no trend. The potential type II error rate is too high to make this statement.
Introduction
Line 48: Add ‘s’ to ‘setting’
Methods
Lines 143-147: It would be beneficial if the author included general information about the tablet and stylus used, and any software or applications to generate the task.
Figure 2: The panel descriptors are reversed. That is, ‘during abduction (bottom: B and D) rather than A and C.
Line 219: The authors mention that AMT was determined with a ‘low level’ contraction. Can the authors provide more detail, e.g., %MVC?
Line 263: Thank you for disclosing the a posteriori decision to analyse writing style.
Lines 264-265: Given the number of participants excluded it would be helpful to provide a statement of the final number included, 16.
Results
Lines 292-298: The authors state that SICI is greater during writing compared to abduction. However, the p value given is 0.052. With such a small sample size, it is unlikely this is a meaningful difference and the type II rate could easily be >35%. The partial eta squared suggests a large effect, the authors need to discuss this in the context that the partial eta squared may overestimate the effect for the population. Hence, this finding is not fully credible as it stands. Although a more significant finding was reported when adding writing styles, this should come with a caveat of caution as it is an a posteriori analysis (bias), and the partial eta squared, again, may be overestimated (small sample and biased measure of population). Post hoc analysis reporting is not complete as the authors report a difference in SICI between printing and abduction in printers (stats provided) but not for the cursive writing group (no stats provided).
Figures 4 & 5: Thank you for including standard deviations as the error and noting that in the figure legend.
Lines 316-328: These results are a little hard to follow. Mostly this arises from not providing the analysis of the data. At least for me, it would be easier to follow if this section laid out main effects and interactions noting the analysis used. It took a fair bit of time to make sure I understood this.
Lines 340-342: Again, the splitting of the groups a posteriori should be noted and as mentioned in general comments, any adjustments that might arise from the violation of sphericity should be considered.
Table 2: It is unclear why the sample sizes do not reflect the reporting in Results based on exclusions.
The authors should include a figure with raw traces for SICI and ICF through the experiment.
Discussion
Lines 412-424: A consideration: Did the authors consider the repetition of the writing task? That is, the actually writing task was the same for all trials. Would there be any difference if the word changed each time and was unknown to the participant?
Lines 425-429: There was no trend. The potential type II error rate is too high to make this statement. The authors might be able to make a case for the effect size, but the statistics need to be much more solid first. Otherwise, remove this discussion point. The authors should clearly declare the potential bias of a posteriori splitting and analysing writing style.
Line 460: Delete ‘in’ between ‘increased’ and ‘facilitation’
Lines 465: Again, a caveat on the a posteriori analysis is needed here.
Author Response
Thank you for the time you have taken to review our manuscript. We appreciate your expertise. I have responded to your comments and queries in-line below. Your comments are in bold, and our responses are in regular font.
Summary: This study examined corticospinal excitability during hand-writing and contrasted this with a common laboratory task before and after fatigue. The authors report that fatigue-induced changes in intracortical facilitation are task dependent, which has implications for studying corticospinal excitability during functional movements outside of the laboratory.
General comments:
Overall, this is a well conducted study with findings that are potentially relevant to those studying cortical excitability and motor control. However, there are a few statistical items that should be clarified. First, given that the actual sample size is 8 (having split the handwriting style into 2 groups), was an a priori power calculation performed? If not, why not? Next, although the authors provide partial eta squared and Cohen’s d, these are not easily interpreted here with such a small sample size. This makes some of the finding uncertain (see specific comments below). Furthermore, with ANOVA with repeated measures, the authors should comment on whether sphericity was tested. If sphericity was violated, an appropriate adjustment needs to be employed (e.g., Greenhouse Geisser correction). For some findings this may change whether the differences reported are significant (see specific comments below) or not.
We started with 20 participants in this repeated-measures study. We based this on a power analysis conducted using data from a previous study from our lab (Sharples et al., PLoS One, 2016) which indicated that we would need 15 participants to detect fatigue-induced differences in SICI, and 20 participants to detect differences in ICF (with α=0.05, β=0.08). The a priori repeated-measures analysis was conducted with 17 participants for SICI (the n was not 8 because hand writing style was not included as a categorical predictor in this a priori analysis, see Table 2).
The analysis of hand writing style (printing vs writing) was not planned in advance. As the study progressed, we found that many university students were unable to write in cursive. From the start, we told participants to write as they preferred given that our aim was to assess corticospinal excitability during a natural motor task. When we found that there were equal numbers of participants who chose to write purely in cursive or purely via printing, we conducted the post hoc analysis. This post hoc analysis with two separate groups of 8 participants is underpowered.
Because we only had two levels for each repeated measure (abduction vs writing, and pre-fatigue vs post-fatigue) we could not test for sphericity (a test that requires three or more levels of each repeated measure). I did consult with others on this suggestion, and so I hope I have not misunderstood your request.
Specific comments:
Abstract
Lines 17-18: There was no trend. The potential type II error rate is too high to make this statement.
The power analysis conducted using data from a previous study from our lab (Sharples et al., PLoS One, 2016) indicated that we would need 15 participants to detect fatigue-induced differences in SICI and 20 participants to detect fatigue-induced changes in ICF (with α=0.05, β=0.08). The repeated-measures analysis of SICI in the present study included 17 participants. However, the current study did differ in several ways from the study upon which we based our sample size calculation, and given the variability observed in the current study, we agree that cautious interpretation is best. We have clarified that this analysis was underpowered in lines 309-312 and included a discussion of this limitation in lines 509-518. We do make a point of referring to the appearance of a trend, because I expect many readers would see the 0.052 and expect to see this reported as a trend. So rather than ignore this perception, we address it by stating, “While there appeared to be a trend toward a greater level of inhibition (SICI) during writing compared to abduction (F(1,16) = 4.40, p = 0.052, = 0.22, Table 2, Figure 5b), it should be noted that this analysis included only 17 participants and was not adequately powered.” (lines 309-312).
Introduction
Line 48: Add ‘s’ to ‘setting’
Thank you. We have corrected this error.
Methods
Lines 143-147: It would be beneficial if the author included general information about the tablet and stylus used, and any software or applications to generate the task.
The task was generated using an iPad. A PowerPoint slide show was used to present a blank 7x2 cm rectangle each time the slide was advanced. The participant was asked to write the word “name” in this box using an Adonit Pro3 precision disc stylus. These details have been added to lines 143-144. In this section, we have also removed reference to “graphic tablet” and instead just referred to the “tablet”.
Figure 2: The panel descriptors are reversed. That is, ‘during abduction (bottom: B and D) rather than A and C.
Thank you very much for catching this significant caption error. We have corrected it.
Line 219: The authors mention that AMT was determined with a ‘low level’ contraction. Can the authors provide more detail, e.g., %MVC?
The procedures used to identify AMT were not correctly described in the manuscript. The criteria for resting motor threshold were mistaken included in the methods. In this study, however, we assessed active motor threshold. AMT was the minimum stimulator output that induced MEPs of more than 200 μV in at least 5 out of 10 consecutive trials during isometric contractions that were matched to the RMS amplitude during hand writing trials conducted during the familiarization phase. As indicated in Table 1, this equates to approximately 20% MVC, a value used to calculate AMT in other papers (e.g. Ni et al., J. Physiol, 583(Pt 3): 971–982). We have corrected this error and provided additional clarification in lines 226-228.
Line 263: Thank you for disclosing the a posteriori decision to analyse writing style.
It was an interesting finding at the end of the study, and although it is underpowered, we did think it worth reporting as it may direct future work.
Lines 264-265: Given the number of participants excluded it would be helpful to provide a statement of the final number included, 16.
The number of participants included in each analysis are included in Tables 2 and 3. We have added a sentence to ensure that readers note this in lines 262-263.
Results
Lines 292-298: The authors state that SICI is greater during writing compared to abduction. However, the p value given is 0.052. With such a small sample size, it is unlikely this is a meaningful difference and the type II rate could easily be >35%. The partial eta squared suggests a large effect, the authors need to discuss this in the context that the partial eta squared may overestimate the effect for the population. Hence, this finding is not fully credible as it stands. Although a more significant finding was reported when adding writing styles, this should come with a caveat of caution as it is an a posteriori analysis (bias), and the partial eta squared, again, may be overestimated (small sample and biased measure of population). Post hoc analysis reporting is not complete as the authors report a difference in SICI between printing and abduction in printers (stats provided) but not for the cursive writing group (no stats provided).
We agree that cautious interpretation is best. We have clarified that this analysis was underpowered in lines 309-312 and included a discussion of this limitation in lines 509-518. We do make a point of referring to the appearance of a trend, because I expect many readers would see the p value of 0.052 and expect to see this reported as a trend. So rather than ignore this perception, we address it by stating, “While there appeared to be a trend toward a greater level of inhibition (SICI) during writing compared to abduction (F(1,16) = 4.40, p = 0.052, = 0.22, Table 2, Figure 5b), it should be noted that this analysis included only 17 participants and was not adequately powered.” (lines 309-312).
The Task*Fatigue*Style ANOVA results are reported in Table 3, and the post hoc results are reported in lines 307-308. The posthoc analysis of abduction vs cursive writing (cursive writers) is reported in 309-310.
Figures 4 & 5: Thank you for including standard deviations as the error and noting that in the figure legend.
We agree that SD is the best measure of variability for these figures.
Lines 316-328: These results are a little hard to follow. Mostly this arises from not providing the analysis of the data. At least for me, it would be easier to follow if this section laid out main effects and interactions noting the analysis used. It took a fair bit of time to make sure I understood this.
We have attempted to clarify this section by describing the statistical tests at the beginning of the paragraph (lines 335-338). If necessary, we could also include a table (as we did for other variables), although we do already have a large quantity of data in tables.
Lines 340-342: Again, the splitting of the groups a posteriori should be noted and as mentioned in general comments, any adjustments that might arise from the violation of sphericity should be considered.
We have clarified the splitting of groups in the Statistical Analysis section (lines 275-282), and have highlighted the limitations of this underpowered a posteriori analysis in the Statistical Analysis section and in lines 509-518 of the Discussion.
Table 2: It is unclear why the sample sizes do not reflect the reporting in Results based on exclusions.
The sample sizes in Table 2 are correct. We recruited 20 participants initially. One was unable to complete the study (as noted in lines 110-111), and SICI data for 2 participants were removed because we were unable to record the MEP during the SICI trials (as noted in lines 260-261). These sample sizes are reflected in the first column of the table. However, the sample sizes were not clear in Figure 5 (previously Figure 4), and this has been revised to more carefully note that the sample size for SICI was 17.
The authors should include a figure with raw traces for SICI and ICF through the experiment.
We have added a figure that illustrates MEP amplitudes and error for one participant before and after fatigue for both the abduction and writing task. This is now Figure 4.
Discussion
Lines 412-424: A consideration: Did the authors consider the repetition of the writing task? That is, the actually writing task was the same for all trials. Would there be any difference if the word changed each time and was unknown to the participant?
This is an interesting question. I expect there would be changes in corticospinal excitability with a task that remained novel rather than repetitive, and that there may have been a more pronounced effect of fatigue, but this is just speculation. We had originally planned to have participants write their signature, a task that we felt would be almost entirely internally-generated, but this was too variable. In the end, we did want to control for changes in muscle activation and hand position, and selected a repetitive task. I do think that the interaction between the physical and cognitive aspects of a written communication task are an interesting future direction.
Lines 425-429: There was no trend. The potential type II error rate is too high to make this statement. The authors might be able to make a case for the effect size, but the statistics need to be much more solid first. Otherwise, remove this discussion point. The authors should clearly declare the potential bias of a posteriori splitting and analysing writing style.
We have clarified the splitting of groups in the Statistical Analysis section (lines 275-282), and have highlighted the limitations of this underpowered a posteriori analysis in the Statistical Analysis section and in lines 509-518 of the Discussion.
Line 460: Delete ‘in’ between ‘increased’ and ‘facilitation’
We have corrected this error.
Lines 465: Again, a caveat on the a posteriori analysis is needed here.
This section is followed by a paragraph that describes the power of the a posteriori analysis as a limitation.
Round 2
Reviewer 1 Report
[Novelty of the study]
(Authors’ response)
R: Although it is well established that corticospinal (CSE) is dependent on voluntary contraction level and task, researchers continue to assess CSE during laboratory tasks that do not reflect natural motor tasks outside the lab. While it is significant that MEP amplitude differs between power and pinch grip, and grasping a dish, and rotating a bottle cap, none of these laboratory tasks reflect the complex natural task of hand writing. Given that this the first study to have assessed corticospinal excitability during hand writing, we do feel that this study is novel.
(Reviewer’s comment)
The difference in the CSE during single joint motor task and multi joint task has been studied. Thus, theoretically, corticospinal activity during writing task must be modulated in accordance with the rules established by the previous findings. Otherwise, we have to assume the contribution of some unknown factors contributing to the control of hand writing. However, even if it is, you can not find the unknown factor on this study design. So, what is the novelty of your study?
This is the only study that reports corticospinal excitability during handwriting; a complex task that is functionally-relevant. Most studies of corticospinal excitability during fatigue use simple movements and the results are extrapolated to functional tasks. We have demonstrated that the effects of fatigue depend on the task used during stimulation. We have assessed some of the factors that might have contributed to this (forearm muscle activity) and controlled others (posture and FDI activity). It was not our aim to determine mechanisms by which CSE are modulated during hand writing. I fail to see how this detracts from the novelty of the study. Our work may direct future studies that explore mechanisms in healthy and clinical populations.
(Author’s response)
In addition, this is the first study to use two different motor tasks during TMS stimulation after exactly the same motor task to elicit fatigue. In so doing, this is the first study to establish that fatigue-associated changes in intracortical excitability depend on the task used during stimulation rather than the task used to fatigue the muscle. This is novel and significant because it addresses highlights the problems associated with extrapolating findings from simple isometric, single-joint operating systems to complex motor tasks outside the laboratory.
(Reviewer's comment)
Rationale of making comparison of the effect of the fatigue on the CSE between the single joint movement and writing is not mentioned. What is the purpose of this comparison? What are you going to know the scientific fact through comparing the effect of fatigue on the CSE between the two tasks? Please clarify.
The rationale is conveyed in the Introduction and explicitly stated in lines 95-98: “Given that measures of CSE depend on limb position, muscle activation level, and other aspects of the motor task, such as different cognitive demands, we speculate that fatigue-induced changes in CSE made during a conventional laboratory task may not represent fatigue-induced changes in CSE made during the more relevant and complex task of writing. Therefore, the purpose of this study was to elicit neuromuscular fatigue and then compare fatigue-associated changes in CSE during writing to the same measures made during a conventional isometric finger abduction task.” In addition to this explicit statement, the first three paragraphs explain that 1) the conclusions drawn from TMS studies depend on the motor task employed during stimulation and may not translate to movement outside the lab (lines 47-48), 2) CSE depends on activation of proximal muscles and task complexity (paragraph 2), and 3) conventional laboratory tasks require different cognitive demands than natural tasks such as writing (paragraph 3).
[Task dependency]
(Authors’s response)
R: Yes, we completely agree that isometric abduction and the natural writing tasks are completely different in many ways. By directly comparing the two in this study, we provide evidence that a simple laboratory task does not yield the same results as a natural motor task. This was our primary objective and the purpose of the study. This study was not designed or intended to determine the factors that contribute to the differences between these tasks. We did control body and arm position with a custom-built apparatus. We also recorded forearm muscle activity because the use of a single joint operating system in conventional laboratory tasks negates the important role that proximal muscle activation may play in setting corticospinal gain. Finally, we did control for muscle activation of the muscle of interest (FDI). In short, this scientific experiment was designed to meet the a priori objectives and test the a priori hypotheses described in the manuscript introduction.
(Reviewer’s comment)
The authors stated that the factors affecting modulation of the CSE is not the purpose of this study. If so, why do you measure the CSE from one muscle (FDI)? I mean that the CSE in one muscle must also be one of the factors that the authors do not mind. What scientific evidence are you going to find through measuring MEP only from one muscle without controlling the other factors? I think if you compare the single joint movement with the natural movement without concerning the multiple different conditions between the tasks, CSE in one muscle is meaningless.
Fatigue studies frequently report data from a single muscle (such as the FDI, the biceps, or the TA). We controlled for FDI muscle activation and report forearm muscle activation. The data are not “meaningless,” as you suggest. Our purpose was to determine “whether changes in CSE in response to a well-studied intervention (neuromuscular fatigue) would depend on the motor task that is used to produce background muscle activity during the delivery of TMS” (lines 86-88) and we hypothesized that, “CSE would be greater during the writing task compared to the simple isometric abduction task…” and that “….that the effect of fatigue on CSE would depend on the motor task employed during stimulation, even when the fatigue task itself was the same.”(lines 101-104).” The experiment was designed to test these hypotheses. Our data support these hypotheses and the notion that studies of fatigue should utilize “experimental paradigms that better reflect relevant motor tasks or at least acknowledge the differences between the task employed in the laboratory and the movements outside of the lab to which results may be extrapolated” (lines 536-538).
[EMG recording]
(Authors’ response)
I have been conducting TMS experiments since 1996, and have never collected TMS data 5000 Hz. In fact, I note that many researchers do not collect at such a high frequency (e.g. Gagné, M., & Schneider, C. (2007). Brain Research; Opie, G. M., Ridding, M.C., Semmler, J. G. (2014). J Neurophysiol; Gordon, C.L., Spivey M. J., Balasubramaniam, R. (2017). Neuroscience Letters; Darling, W. G., Wolf, S. L., & Butler, A. J. (2006). Experimental Brain Research; Kouchtir-devanne, N., Capaday, C., Cassim, F., Derambure, P., & Devanne, H. (2018). J. Neurophysiol.; Bunday, K. L., Tazoe, T., Rothwell, J. C., & Perez, M. A. (2014). Journal of Neuroscience; Hunter, S.K., McNeil, C.J., Butler, J.E., Gandevia, S.C., Taylor, J.L. (2016). Exp Brain Res). Furthermore, Darling et al, (2006) state, "We examined whether sampling at 1,000 Hz was sufficient by recording MEPs in one subject at 5,000 Hz. Regression analysis of peak-to-peak amplitudes (PPamp) of MEPs in the 5,000 Hz sampled data on PPamp from the same data down-sampled to 1,000 Hz (i.e., every 5th point) was used. Correlation coefficients exceeded 0.99 and regression slopes were near 1.0, showing that the 1,000 Hz sampling rate was sufficient to obtain PPamp of EDC MEPs.”
(Reviewer’s comment)
The authors are discussing the sampling rate. I raised concern not only about the sampling rate but also about the high-cut filter of the EMG (450 Hz in this study). I do not know the studies that amplified the MEP with such a low high-cut filter. Please show recent studies that amplified the MEP with high-cut filter with 450 Hz or lower. When measuring the EEG, sampling rate must be twice as much as the frequency of the signal. Thus, in my opinion, the sampling rate to record the EMG is also preferentially greater than twice as much as the high-cut frequency.
The bandpass filter built into our EMG preamplifier (Motion Lab Systems, Inc.) is 15-450 Hz for the FDI. The bandpass filter built into our amplifier (Bagnoli-16, Delsys Inc.) is 20-450Hz for the wrist flexors and extensors. 450Hz is the low-pass cut-off of these standard hardware filters on these commercially available EMG amplifiers. 450 Hz is the low-pass cut-off (the higher end of the band) of this bandpass filter. We are not using a 450-Hz high-pass filter. I agree that that would certainly not make sense.
[TMS timing]
(Authors’ response)
R: I apologize, but I am not sure what you mean by, “…timing of TMS is not cared…”. We agree that timing is very important. As noted in 173-178, timing was tightly controlled. Using frame-based software, we delivered pulses (single or paired) every 5-s. This was true in both the abduction and writing conditions. In the writing condition, the tablet screen (which displayed the command to begin writing) was timed to refresh such that the TMS pulse was delivered on the letters “a” or “m” during the 5-s frame. Any trials that fell between these two letters (between EMG bursts) were excluded. Accordingly, in the writing trials, all stimuli were delivered at exactly 5-s intervals and all MEPs were elicited during either the “a” or the “me” of the written word.
(Reviewer’s response)
I’m sorry for my poor English writing. The participants performed voluntary movement. The time to induce TMS does not matter during tonic abduction of the index finger, but it does during writing. Writing is not a tonic activity. That is, during writing, the strength of the grip and motion direction or velocity of the fingers must phase dependently change. Thus, the experiment must account those phases, and TMS must be time-locked with each movement phase; another words, take the time course for the movement phases during writing.
Yes, we agree. That is why we timed the stimulus to be delivered on the letters “a” or “m”. Because this is a voluntary movement, the timing cannot be timed with any greater precision. As noted, we excluded trials that did not place the stimulus in the EMG bursts for these letters.
(Minor)
Line 208“The ECR and FCR EMG data was”
The word “data” is not singular. “The ECR and FCR EMG data were” may be correct.
Thank you for catching this. We have made this change.